# HAM-ART: An optimised culture-free Hi-C metagenomics pipeline for tracking antimicrobial resistance genes in complex microbial communities

**Lajos Kalmar**[1], **Srishti Gupta**[1], **Iain R. L. Kean**[1], **Xiaoliang Ba**[1], **Nazreen Hadjirin**[1], **Elizabeth M. Lay**[1], **Stefan P. W. de Vries**[1¤a], **Michael Bateman**[1¤b], **Harriet Bartlet**[1], **Juan Hernandez-Garcia**[1], **Alexander W. Tucker**[1], **Olivier Restif**[1], **Mark P. Stevens**[2], **James L. N. Wood**[1], **Duncan J. Maskell**[3], **Andrew J. Grant**[1☯], **Mark A. Holmes**[1☯*]

**1** Department of Veterinary Medicine, University of Cambridge, Cambridge, United Kingdom, **2** The Roslin Institute, University of Edinburgh, Edinburgh, United Kingdom, **3** The University of Melbourne, Victoria Australia

☯ These authors contributed equally to this work.
¤a Current address: MSD Animal Health, Boxmeer, The Netherlands
¤b Current address: Arthur D. Little, New Fetter Place West, London, United Kingdom
* mah1@cam.ac.uk

**Data Availability Statement:** The authors confirm that all data underlying the findings are fully available without restriction. Chromosome

## Abstract

Shotgun metagenomics is a powerful tool to identify antimicrobial resistance (AMR) genes in microbiomes but has the limitation that extrachromosomal DNA, such as plasmids, cannot be linked with the host bacterial chromosome. Here we present a comprehensive laboratory and bioinformatics pipeline HAM-ART (Hi-C Assisted Metagenomics for Antimicrobial Resistance Tracking) optimised for the generation of metagenome-assembled genomes including both chromosomal and extrachromosomal AMR genes. We demonstrate the performance of the pipeline in a study comparing 100 pig faecal microbiomes from low- and high-antimicrobial use pig farms (organic and conventional farms). We found significant differences in the distribution of AMR genes between low- and high-antimicrobial use farms including a plasmid-borne lincosamide resistance gene exclusive to high-antimicrobial use farms in three species of *Lactobacilli*. The bioinformatics pipeline code is available at https://github.com/lkalmar/HAM-ART.

## Author summary

Antimicrobial resistance (AMR) is one of the biggest global health threats humanity is facing. Understanding the emergence and spread of AMR between different bacterial species is crucial for the development of effective countermeasures. In this paper we describe a user-friendly, affordable and comprehensive (laboratory and bioinformatics) workflow that can identify, associate and track AMR genes in bacteria. We demonstrate the efficiency and reliability of the method by comparing 50 faecal microbiomes from pig farms with high-antibiotic use (conventional farms), and 50 faecal microbiomes from pig farms

conformation capture and metagenome sequencing data have been deposited in the European Nucleotide Archive (http://www.ebi.ac.uk/ena) and are available via study accession number PRJEB48382. The complete collection of assembled MAGs is available from the open access Apollo data store at the University of Cambridge (https://doi.org/10.17863/CAM.80312). Underlying numerical data for all graphs and summary statistics is available in the repositories above or in Supporting Information. The code for the HAM-ART pipeline was implemented in Perl programming language. The code and instructions are publicly available under the BSD-3-Clause License at https://github.com/lkalmar/HAM-ART.

**Funding:** This work was funded by the UK Medical Research Council grant MR/N002660/1 to MAH, AJG, MPS, DJM, JLNW, OR. https://mrc.ukri.org The funders had no role in study design, data collection and analysis, decision to publish, or preparation of the manuscript.

**Competing interests:** The authors have declared that no competing interests exist.

with low-antibiotic use (organic farms). Our method provides a novel approach to resistance gene tracking, that also leads to the generation of high quality metagenomic assembled genomes that includes genes on mobile genetic elements, such as plasmids, that would not otherwise be included in these assembled genomes.

## Introduction

The emergence of resistance to antimicrobials in bacteria can occur by spontaneous mutation or by the acquisition of mobile genetic elements carrying antimicrobial resistance (AMR) genes [1] (for example, plasmids *via* natural transformation or conjugation, or bacteriophages *via* transduction [2]). Over the last decade, metagenomic studies have revealed that bacterial communities comprising gut flora or soil microbiota possess a diverse arsenal of AMR genes, termed the resistome [3], some of which can be transferred between related or unrelated species. A limitation of next-generation sequencing metagenomics is the identification of species harbouring a particular AMR gene when that gene is present in extra-chromosomal DNA. Alternative approaches based on traditional culture of bacteria have provided direct experimental evidence of plasmid-mediated AMR gene transfer from enteric pathogens to commensal *Escherichia coli* in rodents [4,5], chickens [6] and humans [7]. *Salmonella*-inflicted enteropathy has been shown to elicit parallel blooms of the pathogen and of resident commensal *E. coli*. These blooms boosted horizontal gene transfer (HGT) in general, and specifically, the transfer of a conjugative colicin-plasmid p2 from an introduced *Salmonella enterica* serovar Typhimurium to commensal *E. coli* [8]. It has been shown that the use of in-feed antimicrobials leads to a bloom in AMR genes in the bacteriophage metagenome recovered from treated pigs [9], although it is unclear what the sources or destinations of these genes are. These observations suggest that HGT between pathogenic and commensal bacteria is a common occurrence in humans and animals and is likely to contribute to the persistence and spread of AMR. Moreover, many previous studies on the spread of AMR from animal sources have focused on AMR of pathogens, with less emphasis on genes within indigenous microbiota that may also pass to humans from animals (and vice versa) but be difficult to culture.

To overcome the inability of next-generation metagenomic sequencing to identify where extra-chromosomal genes of interest reside, several chromosome conformation technologies (such as 3C, Hi-C), originally designed for the study of three-dimensional genome structure in eukaryotes, have been used [10–12]. These techniques exploit the ability to create artificial connections between strands of co-localised DNA by cutting and re-ligating the strands. The techniques differ in their manner of detection, and the scope of interactions they can probe. Marbouty *et al.* describe the application of robust statistical methodology to 3C sequence data (meta3C) derived from a river sediment microbiome [12]. Hi-C, a technical improvement on the 3C method has been shown to successfully disambiguate eukaryotes and prokaryotes [11], and to differentiate closely related *E. coli* strains from microbiomes [10]. Both these techniques offer great potential to define the dynamics of an introduced AMR gene (both chromosomal and extra-chromosomal), in particular the nature and frequency of transfer events, including into microbiota constituents that are not readily detectable by culture in the laboratory. We showcase the performance of an adapted laboratory method using a novel bioinformatic pipeline (HAM-ART), optimised for tracking AMR genes, in a study comparing 100 faecal microbiomes from UK conventional and organic pig farms.

## Results

We adapted exisitng laboratory methodology and developed a bioinformatics pipeline (HAM-ART) that: (i) bins bacterial genomes with high reliability; (ii) associates mobile genetic elements to the host genome; and (iii) annotates and associates AMR genes with high specificity and sensitivity. As HAM-ART is built on traditional metagenomics sequencing methodology, combined with Hi-C sequencing from the same bacterial pellet, it could be applied to any complex microbial community. HAM-ART utilises a widely used sequencing platform, Illumina paired-end sequencing, with standard library sizes and affordable amounts of sequencing per sample. The bioinformatics pipeline was designed to be user friendly, and in addition to generating a set of final metagenomics assembled genomes (MAGs) it outputs results tables reporting assembly quality, taxonomy and AMR gene association.

### Proof-of-concept study undertaken to validate HAM-ART

The HAM-ART methodology was tested in a study comparing AMR in two groups of farms: 5 organic (OG1-5) pig farms, farming to organic certification standards with low antibiotic use, and 5 conventional (CV1-5) pig farms, with higher antibiotic use. Ten faecal samples were taken from each farm for metagenomic analysis as described in the methods section. The organic farms had lower population corrected use (PCU) of antibiotics (average 3.0 mg/PCU, range 0–9.8 mg/PCU) over the year prior to sampling compared to conventional farms (average 85.7 mg/PCU, range 3.9–170.1 mg/PCU). Similarly, the number of different classes of antibiotics used on each farm ranged from 0–4 for organic farms and 4–9 for conventional farms. The results of metagenomic analyses are described below.

### Generation of MAGs using the HAM-ART pipeline

The pipeline produced *de novo* assemblies using approximately 500k contigs from each faecal sample, coupled with 0.2–3.4M informative binary connections from the Hi-C pairs. A Hi-C connection is informative if it connects two different contigs as opposed to a connection within the same contig. The initial products of HAM-ART are the consensus clusters (CCs); a collection of contigs that are clustered together during the network resolution step, solely based on Hi-C contacts, that approximate to a genome of a constituent bacterial species. The total number of CCs for each sample varied between 6k-54k. Of these, we focussed on CCs comprising >250kb (representing about 1/10$^{th}$ of an average prokaryotic genome) for further pipeline processing. After the splitting and extension of the large CCs (as described in the methods section) the number of MAGs varied between 5 and 131 (mean: 62, median: 60) per faecal sample.

A total of 6184 MAGs were identified from the 100 samples which were distributed into 1555 clades based on pairwise genetic distance, indicating groups of MAGs which were likely to represent the same species or genus. The number of members for each clade varied between 1 and 79 (mean: 3.97, median: 2). All clades were subjected to clade refinement that resulted in 553 clades with at least one MAG over 500kb in size. After the clade refinement we ended up with 6164 best quality MAGs.

### Validation of a Hi-C MAG with the matching genome generated from culture of a single isolate

We noted that *E. coli* were relatively rarely assembled in our samples (4% of samples). One possible explanation was that *E. coli* were present, but in low abundance. We investigated this

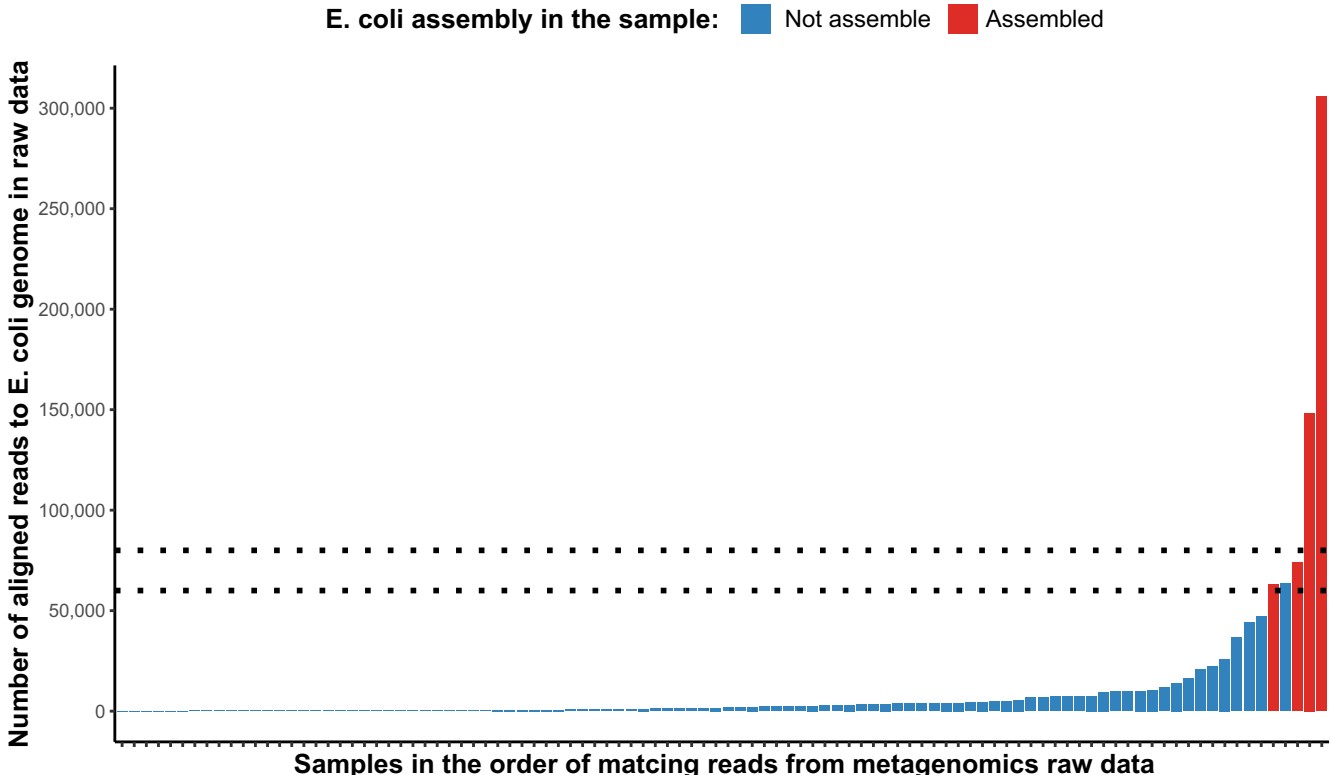

**Fig 1. Presence of *E. coli* DNA in all samples.** Metagenomic shotgun sequencing raw reads from each faecal sample were aligned to a reference *E. coli* genome (*Escherichia coli* O157:H7, GCF_000008865.2) by bowtie2 (—fast option) and the number of reads "aligned concordantly exactly 1 time" were extracted from the output log file. Results were plotted in rank order by the number of aligned reads. In samples plotted as red columns (n = 4) *E. coli* MAGs were successfully assembled using HAM-ART, while those plotted as blue columns were not. Dotted horizontal lines represent the potential threshold range for successful assembly of a MAG (60,000–80,000 reads, representing ~0.2% of the total number of reads for this sample). Repeated analysis using different *E. coli* reference genomes (including an *E. coli* cultured and sequenced from a farm included in this study) gave similar results.

by determining the number of reads in the shotgun libraries from each sample that mapped to an *E. coli* reference genome (Fig 1).

This shows that although reads were present in most shotgun libraries, it was only when there were ≥50k *E. coli* reads that a MAG could be created. This observation suggests that 60-80k reads (representing about 9–12 Mbp), approximating to 2x coverage of an *E. coli* genome are required in order to generate a MAG. In this study, which generated approximately 35M reads (5.25 x$10^9$ bp) for each sample, an individual species would need to represent ~0.2% of the total bacterial community in order to generate a MAG. We believe that the main reason behind the observed sensitivity threshold is the amount of shotgun metagenomics sequencing as we used pooled (and later shared) Hi-C libraries to resolve binning, and the same number of Hi-C contacts were not enough to assemble *E. coli* MAGs if the genome wasn't represented with sufficient coverage.

We examined the quality of a single Hi-C MAG by performing conventional bacterial culture and sequencing of an *E. coli* isolated from the same faecal sample (CV5_05) that generated the *E. coli* MAG. DNA was extracted and the genome obtained using DNA sequencing and assembly (Illumina MiSeq and Spades). The MiSeq data yielded a 5.7 Mbp assembly (CV05-2_S2) that was identified as ST20, and harboured 8 AMR genes (*aadA1*, *aadA2*, *blaCFE-1*, *cmlA1*, *dfrA12*, *mdf(A)*, *sul3* and *tet(34)*). A BLAST comparison of the MiSeq genome with the Hi-C MAG visualised using BRIG is shown in Fig 2.

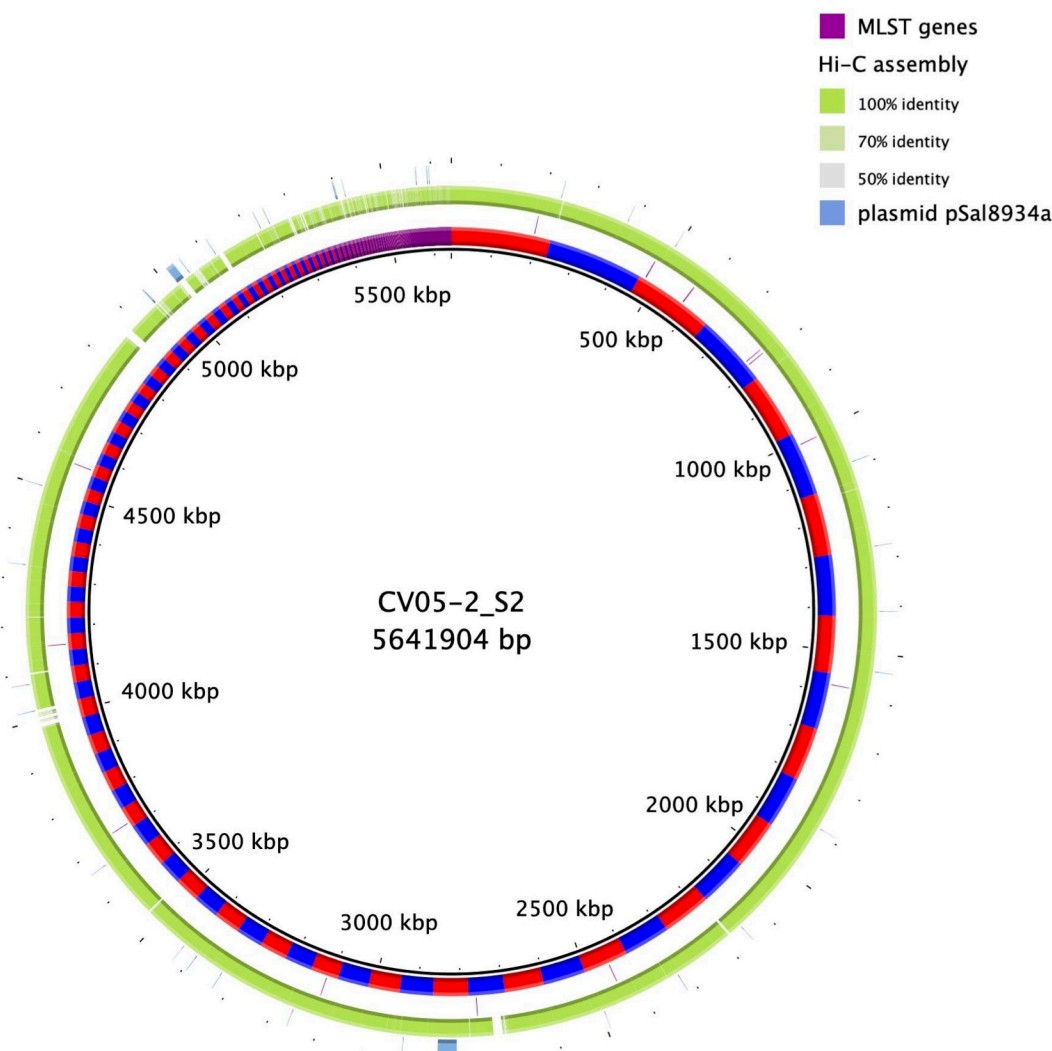

**Fig 2. BLAST comparison of an *E. coli* MAG with a corresponding *E. coli* assembly obtained using culture, followed by Illumina MiSeq sequencing and assembly.** The innermost ring shows the MiSeq assembly with contig boundaries indicated by alternate red and blue colouring. The position of the MLST (Multi-locus Sequence Typing) genes (both MLST schemes 1 and 2) are indicated in the second ring. The matching Hi-C MAG's identity levels are shown in the third ring. The presence of a possible plasmid is illustrated in pale blue in the outermost ring. This ring contains the comparison results for a *S*. Typhimurium plasmid pSal8934a (NCBI accession number JF274993). This plasmid has 99.6% identity and a query coverage of 79% compared to one of the MiSeq assembly contigs (and to the matching MAG). This plasmid contains the *aadA1*, *aadA2*, *cmlA1*, *dfrA12* and *sul3* AMR genes.

## Taxa composition of the pig microbiomes from conventional and organic farms

The distribution of taxa between conventional and organic farms (Figs 3 and S1) were broadly similar apart from OG3. On all farms the diversity included common intestinal bacterial orders, dominated by *Bacteroidales*, *Lachnospirales*, *Lactobacillales*, *Oscillospirales*, which is consistent with previous pig faecal microbiome studies [9,13–15]. The relative paucity of *Enterobacteriaceae* and the presence of a substantial number of treponemes appear to be characteristic of the pig faecal microbiome [16,17].

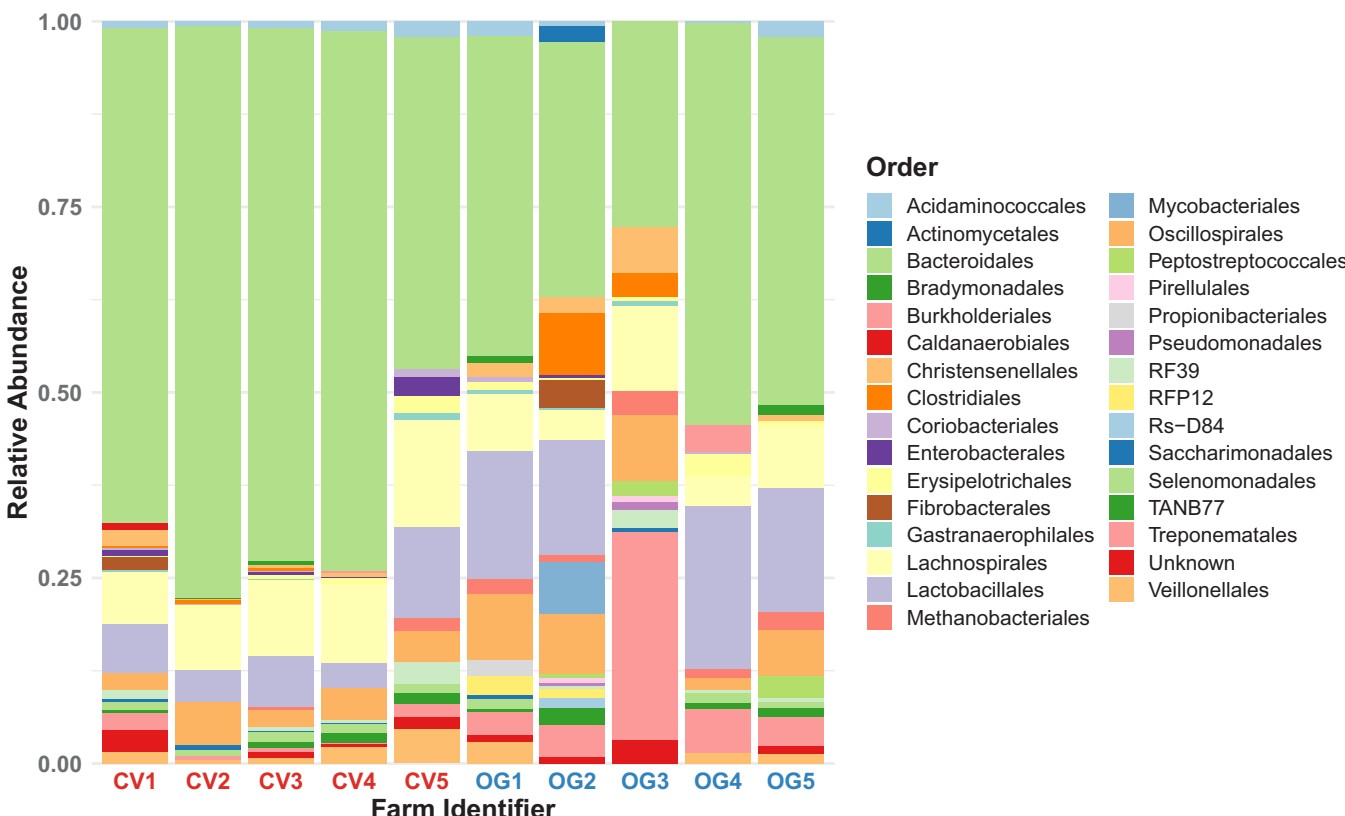

**Fig 3. Order-level relative composition of the pig faecal microbiota in the study farms.** The average relative abundance of different orders identified by GTDB-tk in the final assemblies were calculated from 10 samples in each of the 10 farms included in the study. Plots of different taxonomic levels are shown in S1 Fig.

## AMR gene distribution in faecal samples from CV and OG farms

We identified 66 different AMR genes (in 36 resistance gene groups, as described in methods–analysis of assembly data) using the ResFinder AMR gene database within our final 6164 MAGs (S2 Table). A comparison of the distribution of AMR genes (Fig 4A) indicates that a greater diversity of AMR genes was found in conventional farms compared to organic farms. AMR gene presence/absence was validated by using read-based approach (ARIBA) on the raw sequencing data to ensure that we did not lose any AMR genes due to sensitivity issues during assembly and binning.

Genes encoding proteins potentially able to confer resistance to β-lactams and chloramphenicol were present at greater numbers in samples from conventional farms. A number of genes were present solely in samples from conventional farms (*aac*, *aad*, *blaCMY*, *blaMIR*, *blaOXA*, *blaTEM*, *blaZEG*, *cml*, *dfrA*, *erm(T)*, *lnu(A)*, *mdf(A)*, *mph(N)*, *spc*, *sul*, *van(GXY)*), whereas the gene *vat(E)* was found solely in one ogranic farm. Comparing the number of different AMR genes found in the faecal microbiomes to the antimicrobial use on each farm (using PCU, and the number of different antimicrobials used), we observed statistically significant correlations (Fig 4B and 4C). The correlation of AMR genes with PCU had an R squared value of 71% and P value of 0.0023; the correlation of AMR genes with the number of different antimicrobials used had an R squared value of 80% and P value of 0.0005. While we have a limited number of farms and amount of metadata available from these pig farms, we performed multiple regression analysis on the dataset by using the formula: Number of AMR genes ~

Antimicrobial usage (mg/PCU) + Number of different types of antibiotics used + Farm size (number of pigs) + Organic/Conventional type. Including multiple predictors only slightly increased the correlation coefficient compared to the simple linear regression models (adjusted R squared value of 88%, p = 0.0035).

## Association of *lnu(A)* gene harbouring plasmid to *Lactobacilli* species

An analysis of the distribution of resistance genes among their host MAGs revealed that the lincosamide resistance gene, *lnu(A)*, was found in three clades corresponding to *Lactobacillus amylovorus*, *Lactobacillus johnsonii*, and *Lactobacillus reuteri*. All three clades were present in most samples from both organic and conventional farms, however the *lnu(A)* gene was only present in conventional farms (Figs 4A and 5).

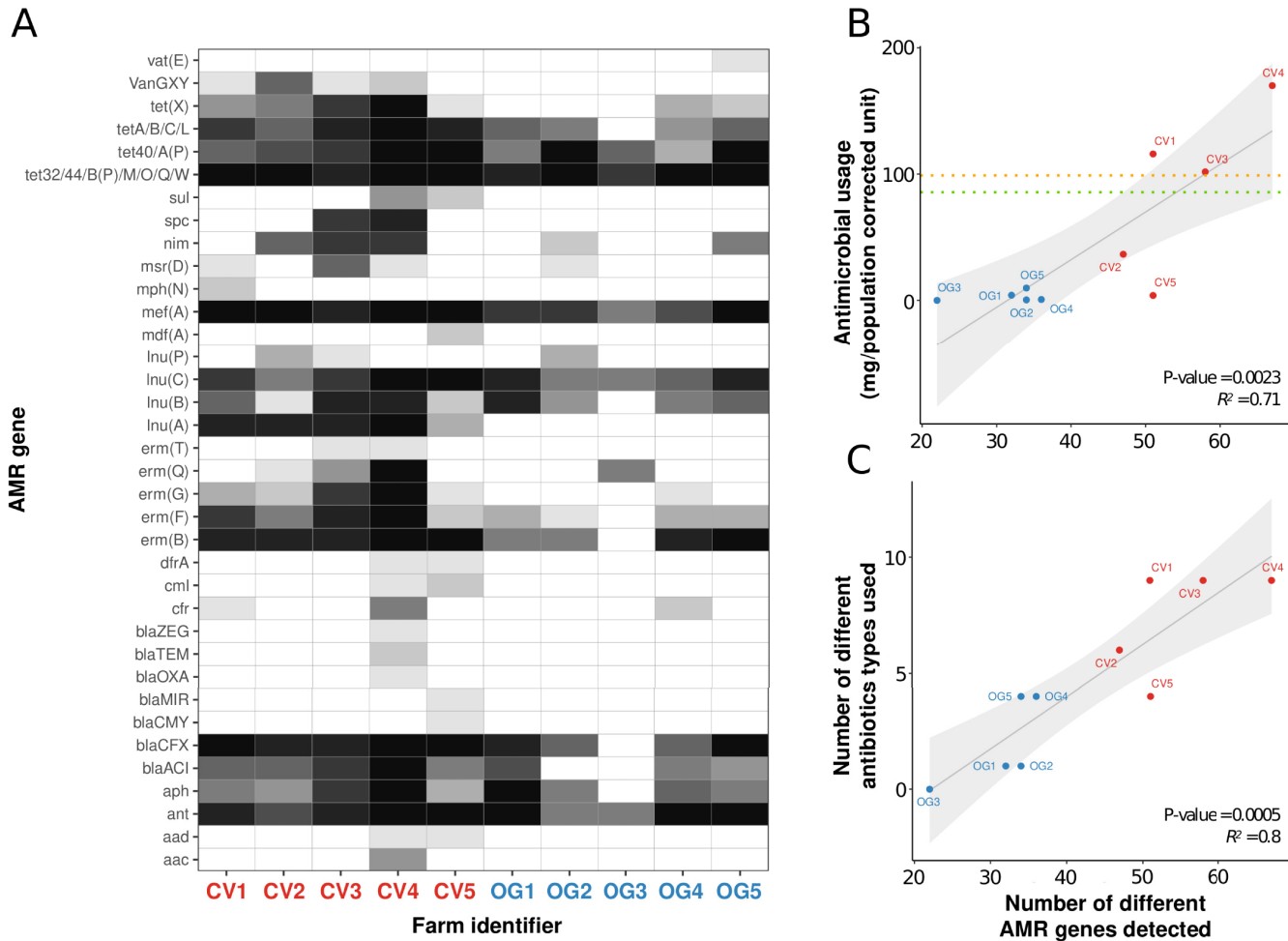

**Fig 4. AMR gene distribution in faecal samples from CV and OG farms and the correlation with levels of antimicrobial used.** Panel A: The heatmap shows the number of samples from which MAGs were generated containing different AMR genes, with the intensity of shading ranging from 0/10 samples (white) to 10/10 samples (black). Conventional (CV1 to 5) and organic (OG1 to 5) are labelled using red and blue text respectively. Panel B: A scatter plot of the amounts of antimicrobial used (mg/population corrected unit (PCU)) in the year prior to sampling, against the number of different AMR genes detected for each farm. The orange line indicates the 2020 target set by the Responsible Use of Medicines in Agriculture Alliance for antibiotic use (99 mg/PCU), and the green line represents the average calculated from the 5 CV farms in our study (85.7 mg/PCU). Panel C: A scatter plot showing the number of different antimicrobial types used in the year prior to sampling, against the number of different AMR genes found. Spearman correlation coefficients and significance values were calculated by fitting a linear regression model to the data points in R (line of best fit and 95% confidence intervals are shaded in grey).

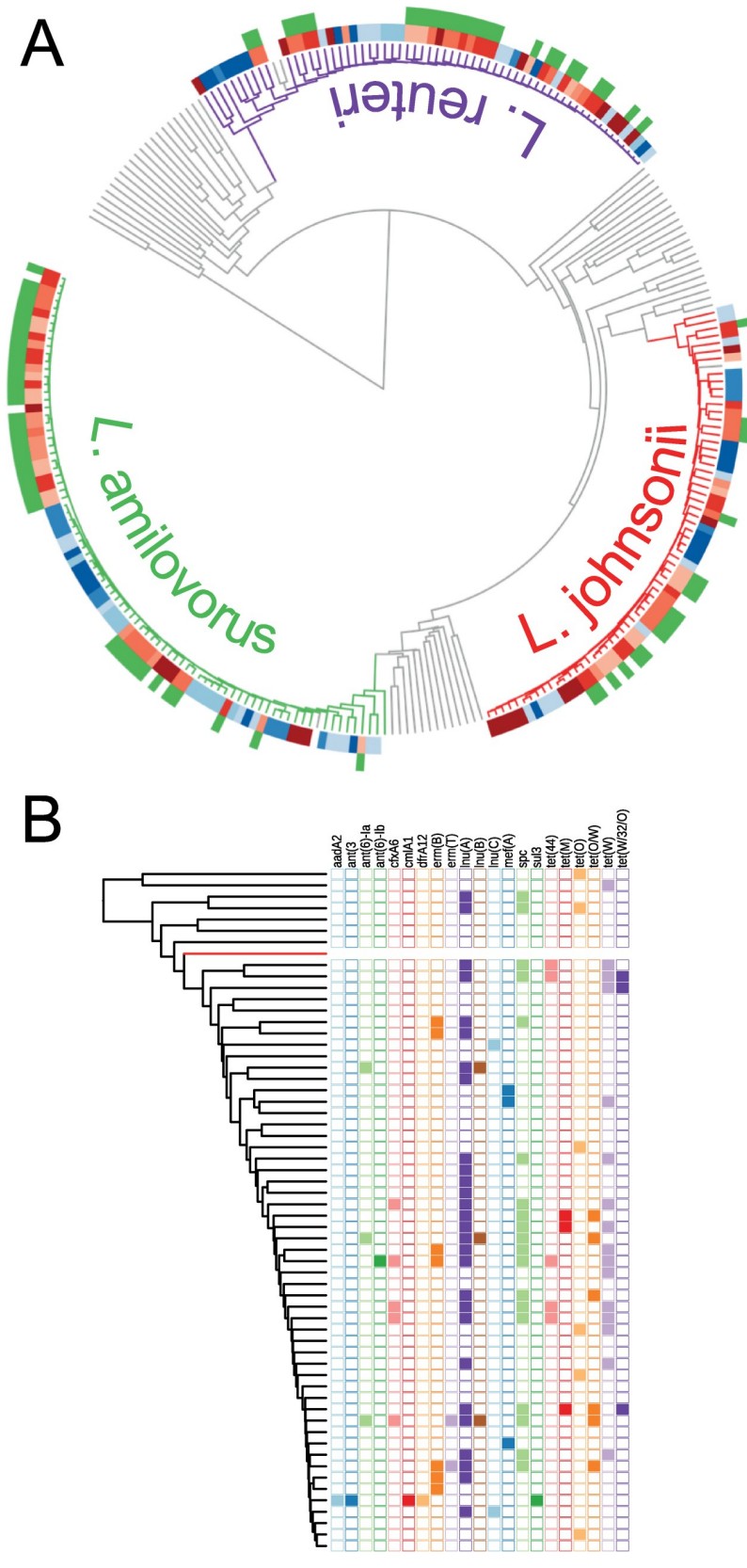

**Fig 5. Association of different AMR genes to *Lactobacilli* species.** Panel A: A distance tree based on sequence comparisons of *L. amylovorus*, *L. johnsonii* and *L. reuteri* assemblies found in farms (coloured branches) together with genomes of the corresponding *Lactobacilli* species (grey branches inside clades) and other known *Lactobacilli* species (grey branches outside the clades) from the NCBI RefSeq collection (https://www.ncbi.nlm.nih.gov/refseq/). The blocks of colour adjacent to the branch-tips indicate the farm type and number (light to dark red: CV1 to 5; light to dark blue: OG1 to 5). The outer circle shows the presence of the *lnu*(A) gene within the MAG (green: present, none: absent). Panel B: A distance tree based on sequence comparisons of all *L. reuteri* MAGs found in the farm samples showing the presence/absence (filled/empty square) of all the AMR genes found in this clade (black branches: MAGs from farm samples, red branch: *L. reuteri* reference genome from NCBI RefSeq collection).

The *lnu*(A) gene was found in all the conventional farms and did not appear to be restricted to a single lineage or species on individual farms. The number of *L. amylovorus*, *L. johnsonii* and *L. reuteri* MAGs obtained from conventional farms were 47, 36 and 42 respectively, and from organic farms were 36, 23, and 25. Examination of the contigs which harboured the *lnu*(A) gene indicated that an identical 5.6 kb sequence was present in 27/34 *lnu*(A) positive *L. amylovorus*, 17/28 *lnu*(A) positive *L. johnsonii* and 18/18 *lnu*(A) positive *L. reuteri* MAGs. The sequence was often present in a single contig of approximately the same length but with different sequencing origins, suggesting that it was present as a circular DNA molecule, potentially a plasmid. In the *lnu*(A) positive lactobacilli MAGs that did not appear to contain the entire sequence, the majority (15/18) had a short contig that was identical to part of the putative plasmid sequence. The plasmid nature of the contig is further strengthened by BLAST search on the NCBI plasmid database. We found that this sequence had 99.8% identity with an 884bp section of a plasmid from a *L. johnsonii* (CP021704) and 83.2% identity with a 1399bp section of a plasmid from a *L. amylovorus* (CP002560).

## Discussion

Conventional shotgun metagenomics sequencing can generate lists of AMR genes and lists of species contained in a microbiome but is not capable of consistently identifying which bacteria carry which plasmid. The application of Hi-C metagenomics in this study demonstrates that this technique can place AMR genes carried on plasmids with their host genomes. The HAM-ART pipeline was tested using a challenging experimental design involving 100 faeces samples from 10 different farms. The results from this study show that it is possible to obtain high resolution, good quality results by performing relatively modest amounts of sequencing on samples of varying quality. While there are other pipelines capable of analysing combined chromosome-capture based assembly and AMR gene association [18–23], during the development of HAM-ART we aimed to design a tool capable of coping with large sample numbers, using the most common Illumina based sequencing platform and delivering results from affordable amounts of sequencing depth.

Unsurprisingly this study shows that farms with lower use of antimicrobials (typically organic farms, who are members of an assurance scheme that strongly regulates the amount of antimicrobials to which the animals are exposed) are associated with smaller numbers and lower diversity of AMR genes, as has been shown in previous studies [24,25]. The statistically significant correlation between the amount of antimicrobial used, and the number of different AMR genes detected for each farm clearly demonstrates this relationship and supports this as a driver of AMR. In future studies collection of further metadata (e.g., main food component ratios, animal density on farms, hygiene rating), may contribute to a more accurate model in predicting the emergence of AMR genes and consequently identify the main factors to increase/decrease. The use of Hi-C metagenomics allows a deeper investigation of the relationship between the use of antimicrobials, AMR genes and the bacteria that harbour those genes.

Of note from this study, is the demonstration of a particular AMR gene, *lnu*(*A*) that was only found in samples from conventional farms. Within conventional farms we found this gene to be harboured in three different species of *Lactobacillus*. All three species of *Lactobacillus* (*L. amylovorus*, *L. Johnsonii* and *L. reuteri*) were also found in organic farms and the distance tree would suggest that similar levels of diversity are present for each species, whether present in an organic or a conventional farm. There is good evidence that the *lnu*(*A*) gene is carried on the same plasmid for all three *Lactobacillus* species, suggesting that any selection pressure selects for the mobile plasmid rather than the host bacteria. The small number of farms, and potential confounders such as geographical bias may have influenced the observed distribution and so this result needs to be confirmed. Nonetheless, the detection of AMR genes, carried on a plasmid, in multiple species without culture could only be performed using chromosome conformation metagenomics techniques such as Hi-C.

A direct assessment of the quality of a Hi-C MAG was afforded by the parallel culture and sequencing of an *E. coli* isolate from the same sample. The homologous Hi-C MAG contained the same MLST and AMR genes as the assembly obtained from conventional culture and sequencing, including AMR genes likely present on a plasmid.

Taxonomic identification of shotgun metagenome assemblies is widely recognised as problematic. We used GTDB-Tk [26], a method using significantly larger genome set than previous algorithms, but were still not able to resolve the taxonomy of many large clades of interest beyond the class level. The chromosome conformation methodology has the potential to generate better quality MAGs by generating links between contigs to improve binning, assembly or even scaffolding. Greater use of Hi-C metagenomics will enable the production of better-quality MAGs for rare or difficult to culture bacteria. Use of the HAM-ART pipeline should also give a lower likelihood of generating mixed or contaminated MAGs.

We performed further investigations to confirm that the use of pooled Hi-C libraries (2 per farm) did not lead to artefactual assembly of MAGs from all 5 of the shotgun libraries that used the same pooled Hi-C library to identify connection pairs. The use of pooled Hi-C libraries reduced costs considerably in terms of staff time, reagents and cost. It is clear from an examination of the distribution of taxa among the samples that there are numerous examples of clades/taxa which we only found in a single sample from a set of 5 sharing the same Hi-C library. Out of our total of 6164 MAGs we would have expected equal distribution between organic and conventional farms but only 2176 came from organic farms and 3988 from conventional farms. While this may have occurred due to a lower species diversity present in the organic farms, it is likely to be a consequence of the larger number of lower yielding Hi-C libraries generated from the organic farms.

The sensitivity threshold for the creation of a MAG from a species contained within a microbiome using Hi-C metagenomic sequencing will be affected by three things. Firstly, the size of the genome of the species of interest (which is likely to be a minor effect); secondly, the amount of sequencing undertaken; and thirdly, the relative abundance of the species of interest, which is probably the most significant influence. The relative abundance of a particular bacterial species may limit the power of the technique when a species of interest may only be present in low numbers. It is likely that there will be some species harbouring AMR genes of interest that are present below a threshold of 1:500 (that we estimate as our theoretical threshold from the *E. coli* content comparison). We used ARIBA to independently assemble AMR genes from our short-read sequencing data and did not find any significant discrepancy between the genes assembled with this method and those found in the MAGs. Indicating that where a gene can be assembled, the Hi-C technique is able to place it in a MAG. We are also aware that Louvain algorithm is not perfect and has its limitations that may introduce slight bias in the HAM-ART pipeline. However, this is one of the most widely accepted network

resolution methods in 3C and Hi-C metagenomics. With more raw data available from Hi-C metagenomics studies, the benchmarking and implementation of other clustering algorithms is likely to be easier in the future. We were unable to perform a direct comparison of different chromosome conformation metagenomics methodology. It is reasonable to observe that our Hi-C libraries in this study do not appear to encode more 3D signal compared to a classical 3C library. The resource costs of performing 3C are lower and for some research questions this approach will be preferable.

In summary, we successfully adapted an existing laboratory method and implemented a bioinformatics Hi-C metagenomics pipeline, HAM-ART, and used it to address a research question using a set of 100 separate samples. We optimised HAM-ART to deal with mixed MAGs, to exploit reference MAGs from within the experimental data set, and to assign AMR genes to the correct MAGs with maximum sensitivity and specificity. While the pipeline focusses on AMR gene tracking for this study, it could be used on other dedicated gene sets (for example a library of virulence-associated genes) to associate these to the host genome. Moreover, it provides a cost-effective strategy to assess the dynamics of AMR transfer longitudinally following treatment with specific antibiotics or doses and following experimental infection. We validated our assembly quality and AMR gene associations by comparing a MAG to one obtained from a cultured *E. coli* from the same faecal sample. We have also shown that the method is robust and affordable when processing large number of samples and provide data illustrating the operational characteristics of both the wet laboratory and bioinformatic protocols involved.

## Methods

### Study population, sampling and data collection

Ethical approval for the sampling and the collection of data was obtained (CR295; University of Cambridge, Department of Veterinary Medicine). All the pig farms sampled were located in southern England and were selected arbitrarily from a list of volunteering farms. The farm descriptors are shown in S1 Table. We sampled five conventional pig farms and five organic pig farms that were members of the Soil Association farm assurance scheme, which stipulates strict controls on the use of antimicrobials. Ten fresh faecal droppings per farm were collected from different groups of fattening pigs aged between 4–20 weeks, transported on ice/cold packs and stored at −80˚C within 6 h of collection. Information on the use of antimicrobials in the one-year period prior to sampling was collected by questionnaire informed by the farm records. The annual use of antimicrobials in mg/number of Population Correction Unit (PCU) was calculated by dividing the total amount of each antibiotic used over the course of a year by the total average liveweight of the animals on the farm considering the numbers of pigs and their ages.

### Enrichment of the microbial fraction from pig faeces

The microbial fraction from a faecal sample was enriched using an adaptation of a previously described method by Ikeda *et al.* [27]. Prior to the enrichment process, 0.5 g of faeces was resuspended in 9 ml of saline and homogenised for 2 min in a Stomacher 80 (Seward) at high power. Debris was removed from the homogenised sample by centrifugation at 500 *g* for 1 min. The supernatant was then transferred on top of 3.5 ml of sterile 80% (w/v) Histodenz (Sigma) and centrifuged in a Beckman ultracentrifuge using a JLA 16.250 rotor at 10,000 *g* for 40 min at 4˚C. After centrifugation, the layer on top of the insoluble debris was recovered into a new 15 ml tube (Falcon) and centrifuged at 500 *g* for 1 min to remove debris. The supernatant was moved to a new 15 ml tube (Falcon) and centrifuged at 10,000 *g* for 20 min at 4˚C.

The bacterial pellet was washed in 10 ml of TE buffer (Merck) and used for the generation of Hi-C libraries.

## Fixation of bacterial cells with formaldehyde

The isolated bacterial fractions from faeces (described in the previous section) were mixed with 2.5% (v/v) formaldehyde (16% methanol-free formaldehyde, Sigma) and incubated at room temperature (RT) for 30 min followed by 30 min at 4˚C to facilitate cross-linking of DNA within each bacterial cell. Formaldehyde was quenched with 0.25 M glycine (Merck) for 5 min at RT followed by 15 min at 4˚C. Fixed cells were collected by centrifugation (10 mins, 10000 rpm, 4˚C) and stored at −80˚C until further use. We pooled bacterial pellets of five samples from the same farm to generate Hi-C libraries, thereby obtaining two Hi-C libraries per farm.

## Generation of Hi-C libraries

The method for the construction of bacterial Hi-C libraries was adapted from Burton *et al*. [11]. Briefly, DNA from the fixed cells was isolated by lysing bacterial pellets in lysozyme (Illumina) followed by mechanical disruption using a Precellys Evolution bead beater (Bertin Technologies, France). Isolated chromatin was split into four aliquots and digested for 3 h at 37˚C using *HpyCH4*IV restriction enzyme (New England Biolabs). Restriction fragment overhangs were filled with biotinylated dCTP (Thermo Scientific) and Klenow (New England Biolabs) as described by van Berkum *et al*. [28]. Biotin labelled digested chromatin was diluted in 8 ml of ligation buffer (New England Biolabs, T4 ligase kit) and proximity ligation was performed at 16˚C for 4 h. De-cross-linking was performed at 65˚C overnight (o/n) with 250 μg/ml proteinase K (QIAGEN). DNA was recovered upon precipitation with 50% (v/v) isopropanol (Fischer Scientific) in the presence of 5% (v/v) 3M sodium acetate (pH 5.2) (Merck) and then treated with RNase A (QIAGEN). Finally, DNA from each sample was recovered in 50 μl TE buffer (Merck) upon phenol-chloroform (Merck) extraction. For Hi-C libraries, biotin from the un-ligated DNA ends was removed by T4 Polymerase (New England Biolabs). DNA was purified using the Monarch PCR and DNA Clean-up Kit (New England Biolabs).

## Generation of Hi-C Illumina sequencing libraries

Illumina sequencing libraries were constructed from purified DNA obtained after Hi-C library preparations using NEBNext Ultra II DNA library prep kit (New England Biolabs). Approximately, 100 ng of DNA of Hi-C libraries was sheared to 400 bp using a Covaris M220 (duty cycle 20%, 200 cycles per burst, peak incident power 50W, treatment time 40 s; Covaris Ltd., UK). Ends of the sheared fragments were repaired, adaptors ligated, and samples were indexed as described in manufacturer's protocols. Before the indexing, we performed semi-quantitative PCR to determine the optimal cycle range for indexing.

## Metagenome sequencing

Metagenomic DNA was isolated from 0.25 g of faeces using Precellys Soil DNA kit (Bertin Technologies, France). Libraries for shotgun metagenome Illumina sequencing were prepared using the NEBNext Ultra II DNA library prep kit (New England Biolabs) upon shearing 250 ng of metagenomic DNA to 400 bp with Covaris M220 (duty cycle 20%, 200 cycles per burst, peak incident power 50W, treatment time 50 s; Covaris Ltd., UK).

### Illumina sequencing of shotgun metagenomic and Hi-C libraries

Following DNA library preparation, the library size was determined with a Bioanalyzer 2100 (Agilent), quantified using the Qubit dsDNA BR kit (Thermo Scientific), pooled appropriately, and analysed with the NEBNext library quant kit (New England Biolabs). The pooled library was subjected to 150 bp paired-end sequencing on the HiSeq 4000 platform (Genomics core facility, Li Ka Shing Centre, University of Cambridge–as 4 shotgun libraries per Illumina HiSeq lane, 1 Hi-C library per Illumina HiSeq lane).

### Bioinformatics pipeline–pre-processing and de-novo assembly

Next generation sequencing raw data files were pre-processed in different ways according to the sequencing library they were derived from. Shotgun metagenomics sequencing data passed through two filtering / quality control steps: (i) optical and PCR duplication removal by using clumpify.sh script from the BBMap software package (https://sourceforge.net/projects/bbmap/ ); (ii) removal of read pairs matching with the host genome using Bowtie2 [29] and the pig reference genome (Sscrofa11.1: GCA_000003025.6). As we performed bacterial cell enrichment during the Hi-C library preparation, we only filtered the raw reads for optical and PCR duplications by using the above-mentioned method. Both raw datasets passed through a merging step, where overlapping (at least 30 nucleotide) reads were merged to one single read, using FLASH software [30]. After merging, metagenomic sequencing reads were passed to the assembly step as paired-end (un-merged) or single-end (merged) reads. All Hi-C sequencing reads were processed further by a Perl script that detected the modified restriction site (in our case A|CGT is modified to ACGCGT) and re-fragmented reads accordingly. This step ensured that hybrid DNA fragments were not used in the assembly step. Re-fragmented Hi-C reads were used in the *de novo* assembly step as single-end reads to increase genome coverages. The pre-processed sequence reads from both libraries were used in *de novo* metagenomic assembly to build-up contigs from overlapping reads by using metaSPAdes [31]. To avoid the introduction of any biases towards known species, we did not use any reference sequence-based assembly method.

### Bioinformatics pipeline–post-processing

Re-fragmented and unmerged Hi-C reads were realigned to the contigs from the assembly by Bowtie2[29] to extract the binary contact information between DNA fragments. The complete list of binary contacts was then transformed to a weighted list and fed into the Louvain algorithm (https://sourceforge.net/projects/louvain/) for 100 iterations of network resolution. Contigs that were clustered together in all 100 iterations were put in the same consensus cluster (CC).

This network resolution method means that a contig can only be assigned to one cluster which may have two unwanted consequences. Firstly, contigs from two or more closely related species may be assigned to the same CC due to sequence homology. The separation of mixed CCs is first addressed using a coverage distribution-based separation algorithm for each CC which splits the CC if the distribution of sequencing coverage was clearly multimodal. In this step we are aiming to extract the core contigs (the longest contig groups with the least variability in coverage) from the mixed CCs, representing potentially most of the genomic DNA. This may easily result in the exclusion of shorter contigs with higher coverage (*e.g.*, multi-copy plasmids) from the CC, but this does not affect the final assembly content. During the next iterative extension step, the potentially connected accessory fragments will be associated again with the CC. The second consequence is that contigs that are shared may not be correctly assigned to all the CCs that should contain copies (*e.g.*, a plasmid possessed by two or more species as a

result of HGT). An iterative CC extension step was built into the pipeline at this point to extend clusters based on the Hi-C inter-contig contacts and cautiously identify contigs that should be allocated to multiple CCs. The detailed workflow of the iterative extension step is summarised in S2 Fig.

Final MAGs were annotated for AMR genes using BLAST [32] using the ResFinder database [33] and taxonomically profiled by GTDB-Tk [26]. AMR genes were also identified from the raw metagenomics sequence reads using ARIBA [34] and compared to the MAG assembly AMR associations to identify the absence of any AMR genes in the final MAGs.

A further clade refinement step in the pipeline exploits the availability of data from multiple samples of the same type (*e.g.*, the other faeces samples from the same study).

## Bioinformatics pipeline–clade refinement

This part of the pipeline undertakes a new assembly iteration using reference genomes from the previous assembly attempt. The main steps of this process were: (i) performing pairwise sequence comparisons between all MAGs (from all samples) by using MASH [35,36]; (ii) using the UPGMA (unweighted pair group method with arithmetic mean) algorithm on pairwise distance data to form clades of closely related MAGs (distance threshold in UPGMA for clade definition: 0.12); (iii) select an exemplar MAG in the clade to use as a within-clade reference sequence; (iv) use the exemplar reference sequence to extract highly similar contigs (using BLAST [32]) from the original full contig collection of the *de novo* assembly for each of the other samples; (v) use Hi-C contact data to refine the collection of contigs extracted by reference search and exclude contigs with no Hi-C contact to other contigs within the MAG; (vi) use Hi-C contacts to extend MAGs with AMR gene containing contigs; (vii) perform a final extension on the MAGs (with the same method as used in the post-processing). We found that the most crucial step during the refinement was the selection of the clade exemplar in the clade that potentially had the most complete genome with minimal contamination. After several attempts of using physical parameters (*e.g.*, using the largest, the median size, the most unimodal coverage distribution) we found that mixed MAGs (mixture of more than one closely related genomes) were also selected as exemplars many times. Therefore, instead of using the above mentioned parameters alone or in combination, we used the core single copy gene set of the GTDB-Tk [26] by running the toolkit "identify" module and looking for: (i) the MAG with the highest number of unique single copy genes (maximum completeness); and (ii) the MAG with the highest unique single copy genes / multiple single copy genes ratio (minimum contamination).

## Bioinformatics pipeline—analysis of MAG sequence data

A set of custom scripts were written to perform AMR gene searches, undertake taxonomic identification, identify closely related reference genomes, and generate paired distance trees for the clades. AMR gene searching was performed using a local installation of BLAST [32] using the ResFinder database [33]. AMR genes were defined as being present where >60% of the length of the target gene was present with an identity of >80%. For AMR gene grouped analysis: (i) the aminoglycoside modifying enzymes were grouped by the modifying group which was attached (aminoglycoside nucleotidyl transferases were grouped together, as were phosphotransferases, acetyltransferases and adenyltransferases); (ii) due to increasing interest in the role ESBL plays in disease, beta-lactamases were grouped by homology; (iii) gene families which were represented by different alleles were considered one gene type; (iv) the dihydrofolate reductase genes *dfrA12* and *dfrA14* are considered as *dfrA*; (v) nitrofuratonin reducing genes were grouped together into the *nim* group; (vi) sulfonamide resistance genes

*sul1-sul3* were grouped as *sul*; (vii) tetracycline resistance genes were grouped by function and sequence homology, with homologous genes combined into groups; and (viii) vancomycin resistance clusters *vanGXY* and *vanG2XY* were considered as one group. Taxonomic identification and the search for closely related reference genomes was performed using GTDB-Tk [26]. Pairwise distances between clade member MAGs and other genomes were determined using MASH [35,36], and the distance tree generated by the UPGMA algorithm. The Newick formatted tree files were annotated using iTOL [37]. Summary text files were automatically created for all clade members with taxonomic identifications and AMR gene associations. A summarised output with all MAGs and AMR gene associations was generated together with a filtered version where incomplete MAGs (filtered out by the default settings of GTDB-Tk [26]) were excluded. For a detailed workflow of the bioinformatics pipeline see S3 Fig.

### Bioinformatics pipeline—quality control

We created custom scripts to extract quality information from almost every step during the pipeline: (i) ratio of duplicated raw reads (detecting low concentration libraries); (ii) ratio of merged raw read pairs (verifying library sizes); (iii) ratio of merged Hi-C reads without detectable ligation site (pointing to problems with Hi-C library preparation); (iv) number of contigs in the *de novo* assembly; (v) Hi-C reads alignment ratio; (vi) number and ratio of informative Hi-C read pairs (a Hi-C read pair is informative if it connects two different contigs); (vii) average modularity during the Louvain network resolution; (viii) single copy gene ratios during clade refinement; (ix) final CheckM-like MAG parameters analysed by GTDB-tk [26] (MAG size, contig number, N50, average coverage, GC-content, taxonomy, completeness). We performed traditional metagenomics assembly on a set (n = 11) of randomly selected samples using the MetaWRAP pipeline (default threshold setting, MetaSpades assembler) [38] and compared the result with the HAM-ART output. While we generally got a higher number of final MAGs from the HAM-ART pipeline (average number of final MAGs 29.5 vs 50.1), due to a few potentially lower quality Hi-C libraries, we had higher variation among the HAM-ART final sets (standard deviation of the mean 13.9 vs 41.2). Quality control information of the 100 (metagenomics) + 20 (Hi-C) sequencing libraries is listed in S3 Table. An additional quality control step was performed on all Hi-C sequencing libraries by using the qc3C tool (pmid: 34634030); results from 1M observations for each library are listed in S4 Table.

### Statistical analysis

Simple linear regressions were performed using R (ggplot2 and ggpmisc packages). Spearman's method was used to determine the P value and correlation coefficient.

### Code availability

The code for the HAM-ART pipeline was implemented in Perl programming language. The code and instructions are publicly available under the BSD-3-Clause License at https://github.com/lkalmar/HAM-ART.

### Supporting information

**S1 Table. Characteristics of the farms used in the study.** The conventional or high-antimicrobial use farms are labelled CV_1 to 5 and the organic, or low antimicrobial use farms are labelled OG_1 to 5.
(PDF)

**S2 Table. Complete list of MAGs and their AMR gene associations from 100 pig faecal samples.** Columns in the tab-separated table are: Farm and sample identification (*e.g.* CV3_2 stands for sample 2 from conventional farm 3); Type of the farm (organic / conventional); Clade identifier; Size of the assembly (in kilobases); Number of contigs in the MAG; N50 of the MAG; Weighted mean coverage of the contigs; GC content of the MAG; GTDB-tk taxonomy string; percentage of the multiple sequence alignment (by GTDB-tk) spanned by the genome; The rest of the columns indicate the presence (1) or absence (0) of the particular AMR gene within the MAG.
(TSV)

**S3 Table. Basic QC information on all the libraries used in this study.** Empty cells and redundant numbers are the indicators of pooled Hi-C libraries.
(XLSX)

**S4 Table. Quality control information on all Hi-C sequencing libraries generated by the qc3C software package.**
(XLSX)

**S1 Fig. Composition of the microbiota on the studied pig farms in domain, phylum, class, order and family levels.**
(EPS)

**S2 Fig. Detailed workflow of the iterative CC and Assembly extension step.** Parameter thresholds are continuously adjusted based on the contig composition of the CC / Assembly.
(TIFF)

**S3 Fig. Detailed bioinformatics pipeline workflow separated to pre-assembly, post-assembly and clade refinement.** Text is coloured black for descriptions and white for the used software / script background.
(PDF)

## Author Contributions

**Conceptualization:** Alexander W. Tucker, Olivier Restif, Mark P. Stevens, Duncan J. Maskell, Andrew J. Grant, Mark A. Holmes.

**Data curation:** Lajos Kalmar.

**Formal analysis:** Lajos Kalmar, Srishti Gupta, Iain R. L. Kean, Stefan P. W. de Vries, Michael Bateman, Olivier Restif, Andrew J. Grant, Mark A. Holmes.

**Funding acquisition:** Alexander W. Tucker, Olivier Restif, Mark P. Stevens, James L. N. Wood, Duncan J. Maskell, Andrew J. Grant, Mark A. Holmes.

**Investigation:** Lajos Kalmar, Iain R. L. Kean, Xiaoliang Ba, Nazreen Hadjirin, Elizabeth M. Lay, Stefan P. W. de Vries, Harriet Bartlet, Juan Hernandez-Garcia, Andrew J. Grant, Mark A. Holmes.

**Methodology:** Lajos Kalmar, Srishti Gupta, Stefan P. W. de Vries, Michael Bateman, Olivier Restif, Andrew J. Grant, Mark A. Holmes.

**Project administration:** Mark P. Stevens, Andrew J. Grant, Mark A. Holmes.

**Resources:** Harriet Bartlet, Juan Hernandez-Garcia.

**Software:** Lajos Kalmar, Michael Bateman.

**Supervision:** Stefan P. W. de Vries, Andrew J. Grant, Mark A. Holmes.

**Validation:** Lajos Kalmar.

**Visualization:** Lajos Kalmar, Srishti Gupta, Iain R. L. Kean.

**Writing – original draft:** Lajos Kalmar, Olivier Restif, Andrew J. Grant, Mark A. Holmes.

**Writing – review & editing:** Lajos Kalmar, Srishti Gupta, Iain R. L. Kean, Xiaoliang Ba, Nazreen Hadjirin, Stefan P. W. de Vries, Michael Bateman, Alexander W. Tucker, Olivier Restif, Mark P. Stevens, James L. N. Wood, Duncan J. Maskell, Andrew J. Grant, Mark A. Holmes.

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
