## [Decision Letter · Decision Letter 0]

13 Sep 2021

Dear Dr Holmes,

Thank you very much for submitting your Research Article entitled 'HAM-ART: An optimised culture-free Hi-C metagenomics pipeline for tracking antimicrobial resistance genes in complex microbial communities.' to PLOS Genetics.

The manuscript was fully evaluated at the editorial level and by independent peer reviewers. The reviewers appreciated the attention to an important problem, but raised some substantial concerns about the current manuscript. Based on the reviews, we will not be able to accept this version of the manuscript, but we would be willing to review a much-revised version. We cannot, of course, promise publication at that time.

If you decide to revise the manuscript for further consideration at PLOS Genetics, please aim to resubmit within the next 60 days, unless it will take extra time to address the concerns of the reviewers, in which case we would appreciate an expected resubmission date by email to plosgenetics@plos.org.

[LINK]

We are sorry that we cannot be more positive about your manuscript at this stage. Please do not hesitate to contact us if you have any concerns or questions.

Yours sincerely,

Xavier Didelot

Associate Editor

PLOS Genetics

Chengqi YI

Section Editor: Methods

PLOS Genetics

Reviewer's Responses to Questions

**Comments to the Authors:**

Reviewer #1: Title: HAM-ART: An optimised culture-free Hi-C metagenomics pipeline for tracking antimicrobial resistance genes in complex microbial communities.

Summary:

This is a particularly well prepared and easy to read manuscript. As a consequence, I have zero minor errors to report. As the authors have discussed, their application of Hi-C to generate MAGs and link AMRs containing sequences is not novel per se. However, their approach to Hi-C MAG refinement that leverages external information sources such as GTDBtk determined single-copy genes, is innovative. At the risk of sounding pompous, I would like to commend them for their determination in pursuing the best possible outcome from their data.

That said, there is sufficient new ground being tread within the work that much more explicit methodological detail is required for each stage (from determining consensus-cluster, splitting and refining). Further, and although I realise this is a tall-order, this reviewer would recommend that a section comprising in-silico validation (possibly against a simulated ground-truth) be added. Lastly, I did find any mention of a code repository for this pipeline, which I would regard as mandatory even at the stage of initial submission. This would have been of great assistance and access to the code would remove part of the burden of describing the method. Altogether, this leaves me with concerns for reproducibility.

Comment:

page 17 line 218: The authors may have found that a read-based approach, which targetted resistance genes, would have had greater sensitivity. This would then modify the following discussion contrasting OG/CV presence/absence. Granted, this would not enable any subsequent gene-context analysis. Note: I see this is briefly addressed at line 370.

page 23 line 313: Although this statement is not in hot dispute for this reviewer, I would have liked there to be a wider consideration of available exogenous variables -- that is to say a more complex regression model. I see that there are some words to this effect further down (line 327) mentioning limitations.

page 24 line 341: Although it provides a taxonomic hint as a side-effect of its validation analysis, the authors of checkM would probably be the first to say that it has never been meant for taxonomic assignment. I would consider revising this sentence.

page 31 line 498: Did the authors consider using or did they trial the use an existing Hi-C metagenomic binning software? Modularity based clustering algorithms (such as Louvain) suffer from a resolution limit which may have impacted the authors

capacity to extract less abundant genomes.

page 31 line 507: This is potentially of interest to the tool developers within Hi-C metagenomics community. As such, more detail is required to describe this method. In particular how do the authors reconcile the variation in coverage that will be seen between core and accessory fragments when splitting?

page 31 line 512: As with my previous comment, this step requires greater detail.

page 32 line 526: The detailed description found in this section, along with the brief descriptions of previous steps, highlights the novelty of the author's approach to refine MAGs. Altogether from splitting to this step, there are a number of methodological points at which things could have gone awry. As such, algorithm validation against a ground-truth would have been greatly appreciated by this reviewer.

Reviewer #2: The manuscript by Kalmar et al. entitled “HAM-ART: An optimized culture-free Hi-C metagenomics pipeline for tracking antimicrobial resistance genes in complex microbial communities” describes a laboratory and a computational pipeline using Hi-C data applied to metagenomic samples. Proximity ligation-based methods (3C, Hi-C) applied to environmental samples is a relatively new field with high potential notably to study mobile genetic elements in microbial communities. In the last years several methods and pipelines have been published in this field demonstrating the increasing interest in this technology. The present manuscript takes advantage of the Hi-C data to study AMR genes and their associated MAGs in a longitudinal cohort of pigs. Not surprisingly, the authors found that a higher antibiotics usage is correlated with a higher AMR genes presence. They also develop and propose a new pipeline to analyze Hi-C data ranging from metagenomic samples.

Even if the field will benefit from new pipelines, the manuscript suffers from various problems that will need to be address before publication.

Major comments:

1 - First, the authors say in the abstract that their pipeline is also a laboratory one but I cannot determine the novelty in the preparation of the libraries compare to previous protocols. Moreover, there is no data to illustrate that their Hi-C data are of good quality in terms of 3D signal. Several tools or calculations exist to check the quality of Hi-C data (3D ratio, q3C) and it will be of great importance to know the quality of their data and to compare them to previous published datasets. Moreover, the authors say that they have implemented several quality controls in their computational pipeline (line 581) but they do not provide them for their libraries.

2- The authors describe succinctly a new computational pipeline but the description and the content will need modifications, descriptions as well as comparison with available ones. First, the authors apply an iterative partitioning procedure using the Louvain algorithm, a procedure and a software already used in previous publications and pipeline. Then they keep group of contigs that cluster together during 100 iterations of the algorithm and they call them Consensus Clusters (CCs); a term very close from the Core Communities (CCs) used in a previous publication and in an available pipeline – MetaTOR (Marbouty et al, 2017, Baudry et al. 2019).

Line 507: “The separation of mixed CCs is first addressed using a coverage distribution-based separation algorithm for each CC which splits the CC if the distribution of sequencing coverage was clearly multimodal.” The authors must provide the algorithm used at this step.

Line 512 :” An iterative CC extension step was built into the pipeline at this point to extend clusters based on the Hi-C inter-contig contacts and cautiously identify contigs that should be allocated to multiple CCs.” Same as previously, details are needed here

If the authors want to publish a new pipeline as they claim it [line 305: “ HAM-ART is the first method that is designed to cope with large sample numbers, using the most common Illumina based sequencing platform and delivering results from affordable amounts of sequencing depth.”], they must provide their code to the community and benchmark their approach with already published pipeline (bin3C, MetaTOR, HicZin …) and other datasets.

3- The paper will be easier to understand if the authors provide a summary table with libraries constructed, assemblies generated, binning statistics … for each sample. Moreover, it will be of great interest for the community to have access to the different assemblies as they can reproduce some of the outputs.

4- The analysis on the lnu(A) gene is interesting but lack several controls and need further analysis. Are the contigs detected as plasmids by state of the art tools like PlasFlow or PlasmidFinder ? Can the authors check the topology of the corresponding contigs by using their Hi-C data ? If the contigs are circular, they should be able to detect the 3D signal in the corresponding contact matrices.

Same remark concerning the E. coli analysis and the possible plasmid.

Minor Comments:

- Line 102 : I would replace “assembles” by “bin” as the pipeline do not assemble the genomes but regroup contigs together.

- In general, there is some problems with the terms “assembled”, “scaffolded” and binned and it is difficult to understand, for instance for the E. coli example, the limits of the method. Is it a problem of assembly (shotgun sequencing) ? a problem of binning (not enough Hi-C reads) ?

- MLST is not defined.

- I have problem to understand exactly how the different assemblies were performed. Do the different Hi-C and Shotgun libraries have been mixed to perform them ? or only the shotgun ones ?

**Have all data underlying the figures and results presented in the manuscript been provided?**

Reviewer #1: Yes

Reviewer #2: **No: **data are not already accessible and i would recommand to also submit the different assemblies generated. Code is also mandatory.

PLOS authors have the option to publish the peer review history of their article (what does this mean?). If published, this will include your full peer review and any attached files.

Reviewer #1: **Yes: **Matthew Z. DeMaere

Reviewer #2: **Yes: **Martial Marbouty

---

## [Decision Letter · Decision Letter 1]

13 Jan 2022

Dear Dr Holmes,

Thank you very much for submitting your Research Article entitled 'HAM-ART: An optimised culture-free Hi-C metagenomics pipeline for tracking antimicrobial resistance genes in complex microbial communities.' to PLOS Genetics.

The manuscript was fully evaluated at the editorial level and by independent peer reviewers. The reviewers appreciated your efforts to account for their previous comments in this revised manuscript. They agreed that your manuscript has improved significantly. One of the reviewers requested a few more minor changes, which we would like you to address in a revised manuscript.

We therefore ask you to modify the manuscript according to the review recommendations. Your revisions should address the specific points made by each reviewer.

[LINK]

Yours sincerely,

Xavier Didelot

Associate Editor

PLOS Genetics

Chengqi YI

Section Editor: Methods

PLOS Genetics

Reviewer's Responses to Questions

**Comments to the Authors:**

Reviewer #1: This reviewer agrees with authors in regards to the on-going difficulty in properly validating new methods in Hi-C metagenomics and I accept that they have tried to address this by sequencing targetted isolates.

Besides this frustration with validation, to which I assign ~no~ blame to the authors, I am satisfied with how the authors have addressed my remarks and appreciate their care in doing so.

Reviewer #2: The revised manuscript by Kalmar appears better than the previous version. Codes and data are now available and would help scientists to reproduce their results. Precisions about their computational pipeline have been also added and their approaches appear promising. The manuscript still need minor modifications / precisions:

1- The authors have provided important statistics about their HiC libraries but never discussed about them. It could of great importance as their HiC libraries do not seem to encode more 3D signal compared to a classical 3C library. Is there a real gain in performing HiC that is more expensive than 3C ?

2- I would really appreciate that the authors submit their assemblies to the NCBI or other databases as it will allow others scientist to reproduce their results without this time-consuming assembly step. It is really easy to do and will help the whole community.

**Have all data underlying the figures and results presented in the manuscript been provided?**

Reviewer #1: Yes

Reviewer #2: Yes

PLOS authors have the option to publish the peer review history of their article (what does this mean?). If published, this will include your full peer review and any attached files.

Reviewer #1: **Yes: **Matthew Z DeMaere

Reviewer #2: **Yes: **Martial Marbouty

---

## [Editor Report · Decision Letter 2]

7 Feb 2022

Dear Dr Holmes,

We are pleased to inform you that your manuscript entitled "HAM-ART: An optimised culture-free Hi-C metagenomics pipeline for tracking antimicrobial resistance genes in complex microbial communities." has been editorially accepted for publication in PLOS Genetics. Congratulations!

Yours sincerely,

Xavier Didelot

Associate Editor

PLOS Genetics

Chengqi YI

Section Editor: Methods

PLOS Genetics

Comments from the reviewers (if applicable):

**Data Deposition**

http://datadryad.org/submit?journalID=pgenetics&manu=PGENETICS-D-21-00617R2

**Press Queries**

---

## [Editor Report · Acceptance letter]

9 Mar 2022

PGENETICS-D-21-00617R2 

HAM-ART: An optimised culture-free Hi-C metagenomics pipeline for tracking antimicrobial resistance genes in complex microbial communities. 

Dear Dr Holmes, 

We are pleased to inform you that your manuscript entitled "HAM-ART: An optimised culture-free Hi-C metagenomics pipeline for tracking antimicrobial resistance genes in complex microbial communities." has been formally accepted for publication in PLOS Genetics! Your manuscript is now with our production department and you will be notified of the publication date in due course.

With kind regards,

Orsolya Voros

PLOS Genetics

On behalf of:
